# Soloxolone *N*-3-(Dimethylamino)propylamide Suppresses Tumor Growth and Mitigates Doxorubicin-Induced Hepatotoxicity in RLS40 Lymphosarcoma-Bearing Mice

**DOI:** 10.3390/ijms262411912

**Published:** 2025-12-10

**Authors:** Arseny D. Moralev, Aleksandra V. Sen’kova, Alina A. Firsova, Daria E. Solomina, Artem D. Rogachev, Oksana V. Salomatina, Nariman F. Salakhutdinov, Marina A. Zenkova, Andrey V. Markov

**Affiliations:** 1Institute of Chemical Biology and Fundamental Medicine, Siberian Branch of the Russian Academy of Sciences, Lavrent’ev Ave., 8, 630090 Novosibirsk, Russia; arseniimoralev@gmail.com (A.D.M.); senkova_av@1bio.ru (A.V.S.); ana@nioch.nsc.ru (O.V.S.); marzen@niboch.nsc.ru (M.A.Z.); 2Faculty of Natural Sciences, Novosibirsk State University, Pirogov St. 2, 630090 Novosibirsk, Russia; alina.okhina@mail.ru (A.A.F.); d.solomina@g.nsu.ru (D.E.S.); rogachev@nioch.nsc.ru (A.D.R.); 3N. N. Vorozhtsov Novosibirsk Institute of Organic Chemistry, Siberian Branch of the Russian Academy of Sciences, Lavrent’ev Ave., 9, 630090 Novosibirsk, Russia; anvar@nioch.nsc.ru

**Keywords:** soloxolone, doxorubicin, glycyrrhetinic acid, hepatoprotection, lymphosarcoma, multidrug resistance, NRF2 signaling, oxidative stress, P-glycoprotein

## Abstract

Multidrug resistance (MDR) remains a significant obstacle to effective cancer chemotherapy, primarily due to overexpression of P-glycoprotein (P-gp), which reduces intracellular accumulation of cytotoxic drugs. This study evaluated the pharmacological potential of the glycyrrhetinic acid derivative soloxolone N-3-(dimethylamino)propylamide (Sol-DMAP) as a biocompatible P-gp inhibitor with hepatoprotective properties. Using a murine model of P-gp-overexpressing RLS40 lymphosarcoma, we demonstrated that Sol-DMAP significantly enhanced the antitumor efficacy of doxorubicin (DOX) by increasing its intratumoral concentration 4.7-fold without enhancing systemic toxicity. Independently, Sol-DMAP exhibited direct antitumor activity, reducing tumor growth in vivo and inducing apoptosis and G1-phase arrest in RLS40 cells in vitro. In addition, Sol-DMAP mitigated DOX-induced hepatic injury by reducing necrotic and dystrophic changes in liver tissue and restoring heme oxygenase 1 (Hmox1) expression. Further studies in HepG2 cells confirmed that Sol-DMAP activated the NRF2-dependent antioxidant response, upregulating *HMOX1*, *GCLC*, *GCLM*, and *NQO1* genes. Molecular docking revealed that Sol-DMAP can disrupt the KEAP1-NRF2 interaction, likely leading to NRF2 activation. Collectively, these findings demonstrate that Sol-DMAP effectively reverses P-gp-mediated MDR while protecting the liver from oxidative stress, highlighting its potential as a multifunctional scaffold for the development of safer and more effective chemotherapeutic adjuvants.

## 1. Introduction

The emergence of multidrug resistance (MDR) remains a major obstacle in cancer chemotherapy, limiting the efficacy of antitumor drugs and often resulting in treatment failure and disease recurrence [1]. Tumor cells can acquire resistance through various mechanisms [2]. The most common of these is the increased efflux of cytotoxic agents mediated by ATP-binding cassette (ABC) family transporters [3,4]. P-glycoprotein (P-gp; also known as MDR1 and ABCB1), the most extensively studied ABC member, actively pumps out a wide range of chemotherapeutics (e.g., doxorubicin (DOX), paclitaxel, vincristine) out of tumor cells, thereby reducing their therapeutic efficacy [5,6] and, thus, is considered as a valuable pharmacological target for overcoming MDR [7,8,9].

Despite promising in vitro efficacy, clinical trials of P-gp inhibitors have been largely unsuccessful due to their low MDR-reversal potency and off-target toxicity [10]. The overall failure of these compounds is primarily because P-gp is physiologically expressed in normal organs, such as the liver, kidney, gastrointestinal tract, and blood–brain barrier, where it functions in xenobiotic clearance and tissue protection [11]. Consequently, first- and second-generation P-gp inhibitors (e.g., verapamil and cyclosporine A; Figure 1), which have low selectivity, demonstrate pronounced systemic toxicity [12,13,14]. More specific third-generation agents like tariquidar, elacridar, and zosuquidar (Figure 1) still failed to demonstrate consistent therapeutic benefit in clinical trials, partly due to altered pharmacokinetics and toxicity profiles [10]. These challenges highlight the need for the continued development of novel P-gp inhibitors that demonstrate not only high inhibitory potency but also protective properties for organs and tissues constantly expressing P-gp.

A significant concern during therapeutic intervention with P-gp inhibitors is the state of the liver, which is a central organ for xenobiotic metabolism. Various antitumor drugs, such as DOX, cisplatin, and paclitaxel, in addition to their primary effect on tumor cells, induce excessive generation of reactive oxygen species (ROS) in hepatocytes. This leads to oxidative stress–mediated damage and, consequently, pronounced hepatotoxicity [15,16,17]. The use of P-gp inhibitors can seriously potentiate this effect. Indeed, tariquidar and valspodar have been reported to increase the accumulation of paclitaxel and sunitinib, respectively, in liver tissue in vivo [18,19,20]. Kong et al. demonstrated that co-administration of tariquidar enhanced triptolide-induced hepatotoxicity in mice, as evidenced by pathological liver changes and elevated serum levels of alanine aminotransferase (ALT) and aspartate aminotransferase (AST) [21]. In accordance with this, significant enhancement of trabectedin-induced hepatotoxicity was observed in MDR1a/1b knockout mice [22]. Thus, the development of P-gp-targeting compounds that can mitigate hepatic injury caused by an increased xenobiotic load represents a promising direction in drug development.

Given its proven hepatotropic effect [23,24] and multitargeting profile [25,26], 18βH-glycyrrhetinic acid (GA) represents an important structural platform for creating novel P-gp inhibitors with optimized pharmacological properties. We previously synthesized a semisynthetic GA derivative, soloxolone *N*-3-(dimethylamino)propylamide (Sol-DMAP, Figure 1), which demonstrated a high inhibitory effect on P-gp efflux function in P-gp-overexpressing human cervical carcinoma KB-8-5 and murine lymphosarcoma RLS40 cells [27]. We showed that Sol-DMAP effectively restores the sensitivity of these cells to DOX due to its direct interaction with the P-gp transmembrane domain [27]. Despite its pronounced anti-P-gp potency, the MDR-reversal activity of Sol-DMAP in vivo has not yet been verified.

The present study aimed to address three key questions essential for a deeper understanding of the pharmacological potential of Sol-DMAP: (i) whether it enhances the antitumor efficacy of DOX in a P-gp–overexpressing RLS40 lymphosarcoma murine model, (ii) whether this triterpenoid possesses its own antitumor activity in vivo, and (iii) whether it exerts a protective effect against DOX-induced hepatic toxicity in experimental mice. Using an integrated pharmacological, histological, biochemical, and computational approach, we demonstrate that Sol-DMAP exerts a complex inhibitory effect on RLS40 lymphosarcoma growth, which is accompanied by enhanced sensitivity to DOX treatment and activation of antioxidant defense mechanisms in liver tissue. Collectively, these results highlight Sol-DMAP as a promising biocompatible P-gp inhibitor and underscore the high potential of GA as a noteworthy multifunctional scaffold for safe and effective MDR reversal.

## 2. Results

### 2.1. Sol-DMAP Potentiates DOX and Exhibits Direct Antitumor Activity in Murine RLS40 Lymphosarcoma In Vivo

#### 2.1.1. Sol-DMAP Enhances the Antitumor Efficacy of DOX by Increasing Its Intratumoral Concentration

Based on the previously demonstrated anti-P-gp activity of Sol-DMAP in RLS40 cells in vitro [27], this study utilized a murine model of intramuscularly implanted RLS40 lymphosarcoma [28,29].

Initially, to determine the optimal timing for Sol-DMAP and DOX injections in the subsequent experiments, the pharmacokinetic profile of Sol-DMAP was analyzed. Intact mice were injected intraperitoneally with the triterpenoid at a dose of 20 mg/kg, followed by blood collection at various time points and quantification of Sol-DMAP concentration using high-performance liquid chromatography–tandem mass spectrometry (HPLC–MS/MS). As shown in Figure 2A, the pharmacokinetic profile of Sol-DMAP in blood is characterized by a rapid increase in its concentration between 15 min and 1 h, reaching a maximum of 2000–2250 ng/mL at 0.5–1 h post-injection, followed by a gradual decline to baseline level by 24 h.

Based on the obtained pharmacokinetic data, the following experiments were designed with DOX administered 1 h after Sol-DMAP injection to achieve optimal potentiating effects. The experimental setup is depicted in Figure 2B. Mice bearing RLS40 lymphosarcoma were treated intraperitoneally with Sol-DMAP (20 mg/kg), DOX (5 mg/kg), or their combination applied in the same doses three times per week, yielding a total of five administrations. The experiment was terminated on day 14 after tumor implantation (Figure 2B).

Analysis of tumor volume revealed that, starting from day 6, Sol-DMAP enhanced the antitumor efficacy of DOX. The most pronounced effect was observed on day 8, where the combination treatment exceeded the effects of Sol-DMAP and DOX monotherapies by 2.6- and 2.5-fold, respectively (Figure 2C). Notably, the potentiating action of Sol-DMAP diminished by the end of the experiment. On day 13 post-tumor implantation, the antitumor effect of the combined regimen was no longer different from that of DOX administered alone, with both treatments resulting in 2.2- and 1.9-fold reductions in tumor volume compared to the vehicle group, respectively (Figure 2C). Interestingly, Sol-DMAP administered alone also demonstrated a direct antitumor effect, reducing tumor volume by 57% on day 11 compared to the vehicle group (Figure 2C). Similarly to DOX desensitization in the combined regimen, the inhibitory effect of Sol-DMAP on tumor growth weakened by the end of the experiment. This may indicate a significant influence of rapid tumor progression on the efficacy of Sol-DMAP and highlights the need for further optimization of the treatment regimen. Despite this, we confirmed the ability of Sol-DMAP to potently, albeit not persistently, potentiate the antitumor effect of DOX in RLS40 lymphosarcoma in vivo.

Next, to determine whether the observed Sol-DMAP-induced enhancement of DOX efficacy was associated with its P-gp-inhibiting activity or was merely due to synergistic tumor-suppressing effects, the in vivo experiment was repeated (Figure 2B). Tumor nodules from mice injected with DOX or Sol-DMAP+DOX were isolated on day 8 post-tumor inoculation and analyzed by HPLC-MS/MS. It was shown that a course of three Sol-DMAP administrations (20 mg/kg) led to its effective accumulation in tumor tissue at 610 ng/g (Figure 2D) and a 4.7-fold increase in the intratumoral concentration of DOX compared with the DOX-only group (Figure 2E). These findings confirm that Sol-DMAP sensitizes RLS40 tumors to DOX by inhibiting P-gp-mediated efflux, thereby enhancing DOX retention and efficacy in vivo.

#### 2.1.2. Sol-DMAP Is Well-Tolerated in RLS40 Lymphosarcoma-Bearing Mice

Body weight monitoring revealed a 12% decrease in mice injected with DOX (Figure 2F), which is consistent with its known ROS-dependent systemic toxicity [30]. Notably, the administration of Sol-DMAP in the combination regimen significantly diminished DOX-induced body-weight loss, suggesting a possible protective effect of the investigated triterpenoid. The obtained results demonstrated that Sol-DMAP administered alone or in combination with DOX is well-tolerated in vivo, as its five-dose regimen did not cause a reduction in animal body weight (Figure 2F; Appendix A) or affect liver and kidney organ indices (Figure 2G).

#### 2.1.3. Sol-DMAP Decreases Necrotic Area and Mitotic Activity in RLS40 Lymphosarcoma Tissue

To further explore DOX-potentiating and antitumor effects of Sol-DMAP at the final stage of the experiment, a histological and immunohistochemical analysis of tumor nodes collected on day 14 was performed. Histologically, RLS40 lymphosarcoma in the control and vehicle-treated groups was represented by polymorphic lymphoid cells with high mitotic activity and necrotic decay in the central areas of tumor nodes (Figure 3A–C). Sol-DMAP treatment altered the histological characteristics of RLS40 lymphosarcoma, notably by reducing the necrosis expansion and mitotic activity in the tumor nodes. The volume density of necrotic areas in Sol-DMAP-treated tumors was 2.1-fold lower compared to the vehicle group, while combination therapy led to a 1.6-fold decrease in this parameter compared to the DOX-alone group (Figure 3A, upper panel; Figure 3B).

Regarding the proliferative potential of the tumor, all experimental groups exhibited a significant decrease in the mitotic activity of RLS40 cells. This was evidenced by reduced numerical densities of the cells in mitosis by 2.8-, 4.9-, and 7.6-fold in the DOX, Sol-DMAP, and combination therapy groups, respectively, compared to the vehicle group (Figure 3A, middle panel; Figure 3C). These histological findings confirm the pronounced anti-tumor activity of Sol-DMAP. Despite the statistically non-significant difference in tumor volume between the Sol-DMAP and vehicle groups observed on day 14 (Figure 2C), the triterpenoid continued to effectively inhibit the mitotic activity of lymphosarcoma cells until the end of the experiment (Figure 3C).

Further semiquantitative assessment of the intensity of P-gp immunohistochemical staining in tumor sections revealed that treatment with Sol-DMAP, either as a single agent or in combination with DOX, resulted in an approximately 2-fold decrease in P-gp expression, while DOX alone did not affect P-gp level (Figure 3D). Additional RT-qPCR analysis also demonstrated a reduction in *Mdr1b* expression following Sol-DMAP treatment; however, these changes were not statistically significant (Figure 3E). Collectively, these results indicate that Sol-DMAP can not only directly inhibit P-gp pump function [27] but also downregulate P-gp expression in the late stages of RLS40 lymphosarcoma growth in vivo (Figure 3A,D).

#### 2.1.4. Sol-DMAP Induces ROS-Independent Cell Death via Apoptosis and G1-Arrest in RLS40 Cells In Vitro

Given the observed direct inhibitory effect of Sol-DMAP on RLS40 lymphosarcoma growth, we asked which mechanism underlies this activity. Consistent with the in vivo data (Figure 2C), Sol-DMAP exhibited a cytotoxic effect in RLS40 cells. Interestingly, the explored triterpenoid showed a 3.7-fold higher potency than DOX (IC_50_ values of 35.2 µM and 131.8 µM, respectively; Figure 4A). In contrast, Sol-DMAP was less cytotoxic than DOX against non-malignant immortalized cells, including human embryonic kidney HEK293 cells, Madin-Darby canine kidney MDCK cells, and murine J774 macrophages. The IC_50_ for Sol-DMAP in these cell lines ranged from 11.7 to 18.4 µM, compared to 2.3–4.0 µM for DOX (Appendix A). We hypothesize that this difference is related to the MDR phenotype of RLS40 cells. The reduced sensitivity of non-malignant cells to Sol-DMAP compared to DOX (Appendix A) suggests a potentially favorable safety profile for the triterpenoid, which is consistent with the good tolerability of Sol-DMAP observed in vivo (Figure 2F,G). As in the mouse model, Sol-DMAP significantly potentiated the cytotoxicity of DOX in RLS40 cells, reducing its IC_50_ value by 3.4-fold in the combination regimen compared to DOX alone (Figure 2C, insect).

To elucidate the mechanism of cell death induced by Sol-DMAP and its combination with DOX, RLS40 cells were treated with the specified compounds for 24 h and stained with Annexin V-FITC/propidium iodide (PI). As shown in Figure 4B,C, Sol-DMAP markedly increased the population of late apoptotic cells compared to the control. Interestingly, DOX primarily induced early apoptosis in RLS40 cells, whereas Sol-DMAP significantly potentiated its pro-apoptotic activity, triggering late apoptosis in 96.5% of cells in the Sol-DMAP+DOX group (Figure 4B,C). None of the treatments led to a significant accumulation of necrotic cells (Figure 4B, upper left quadrants).

The revealed pro-apoptotic effect of Sol-DMAP was independently confirmed by measuring caspases-3/-7 activity. As depicted in Figure 4D, a 6 h incubation of RLS40 cells with Sol-DMAP, corresponding to an early apoptotic stage, increased caspases-3/-7 activity by 24% compared to the control. DOX demonstrated a slightly more pronounced effect (33.3%); however, its combination with Sol-DMAP did not lead to a further increase in caspase-3/-7 activation (Figure 4D).

Given that pentacyclic triterpenoids are known to trigger tumor cell death through ROS production [31,32], we next evaluated whether the cytotoxicity of Sol-DMAP was ROS-dependent. Pretreatment of RLS40 cells with the ROS scavenger N-acetyl-L-cysteine (NAC) did not alter the cytotoxic profile of Sol-DMAP (Figure 4E). Furthermore, DCFDA-based ROS detection revealed no significant difference in the intracellular redox state between control and Sol-DMAP-treated cells, indicating that the triterpenoid acts through a ROS-independent mechanism (Figure 4F). In contrast, DOX significantly increased ROS levels in RLS40 cells. Notably, the combination of Sol-DMAP with DOX completely abrogated the ROS-stimulating activity of DOX (Figure 4F). The absence of a ROS response to Sol-DMAP alone in RLS40 cells appears to be cell-context dependent, as an additional experiment showed that Sol-DMAP increased ROS production by 1.5-fold in J774 macrophages (Appendix A).

In light of the pronounced anti-mitotic activity of Sol-DMAP observed in RLS40 tumor tissue (Figure 3C), we investigated whether this compound could affect the cell cycle. Flow cytometry analysis of DAPI-stained RLS40 cells revealed that Sol-DMAP induced G1-phase arrest, increasing the G1 population by 2.1-fold, while reducing the S- and G2-populations by 1.4- and 2.0-fold, respectively, compared to the control (Figure 4G,H). Interestingly, while DOX induced a G2/M arrest in RLS40 cells, the combination treatment was dominated by the effect of Sol-DMAP, shifting the cell cycle profile to a G1-phase arrest (Figure 4G,H).

Collectively, these results demonstrate that the antitumor effect of Sol-DMAP is ROS-independent and mediated by the induction of caspase-dependent apoptosis and G1-phase arrest in RLS40 cells.

### 2.2. Sol-DMAP Demonstrates a Hepatoprotective Effect in DOX-Treated Mice

#### 2.2.1. Sol-DMAP Attenuates DOX-Induced Liver Damage in RLS40 Tumor-Bearing Mice

Having established the DOX-potentiating and antitumor activities of Sol-DMAP, and considering the hepatotropic properties of GA and its derivatives [23], we next investigated the modulating effect of Sol-DMAP on DOX-induced hepatic injury in RLS40 tumor-bearing mice (Table 1, Figure 5).

Histological analysis of the livers revealed that tumor growth caused a manifold increase in destructive changes, presented by the dystrophy and necrosis of hepatocytes (Table 1, Figure 5A, left panel). The administration of DOX to some extent reduced the intensity of these changes, however in parallel led to 3.6- and 2.5-fold increase in the blood vessels component in the structure of liver tissue compared to healthy and control tumor-bearing animals, respectively, indicating a general toxic and, in particular, cardiotoxic effect of DOX (Table 1). Treatment with Sol-DMAP alone reduced destructive signs in the liver parenchyma of RLS40-bearing mice, primarily through a 2.2- and 1.6-fold reduction in dystrophies compared to the control and vehicle-treated groups, with no significant effect on necrosis (Table 1). The most pronounced protective effect on the liver was observed with combination therapy, which reduced both dystrophic and necrotic alterations. This resulted in a 2.4- and 2.1-fold decrease in total destructive changes in the liver tissue compared to the control and vehicle-treated mice, respectively (Table 1). Notably, Sol-DMAP effectively normalized the DOX-damaged liver vasculature; the volume density of blood vessels was 2.7-fold lower in the Sol-DMAP+DOX-treated group than in the DOX-alone group (Table 1).

Considering the key regulatory function of heme oxygenase 1 (Hmox1) in hepatoprotection [33] and the proven Hmox1-stimulating properties of soloxolone methyl, a structural analog of Sol-DMAP [34,35], we further evaluated the hepatic Hmox1 level in control and experimental mice. Immunohistochemical staining confirmed a compensatory increase in Hmox1 levels in the liver tissue of RLS40 lymphosarcoma-bearing mice from the control and vehicle-treated groups, which we attribute to the tumor-associated liver damage (Figure 5A, right panel). Sol-DMAP alone did not affect Hmox1 expression in the liver, whereas DOX caused a dramatic decline in its expression. Notably, the addition of Sol-DMAP to DOX treatment to a certain extent prevented the DOX-induced downregulation of Hmox1 in the liver tissue, confirming the hepatoprotective effect of the investigated triterpenoid.

#### 2.2.2. Sol-DMAP Induced Antioxidant Response in Hepatocyte-Like Cells In Vitro

Since DOX is known to cause hepatotoxicity primarily through oxidative stress induction [30], we asked whether hepatoprotective effect of Sol-DMAP observed in vivo can be associated with its antioxidant potential.

Initially, the structural similarity analysis revealed that Sol-DMAP shares key features with oleanolic acid derivatives shown in Figure 5B, such as 2-cyano-3,12-dioxooleana-1,9-dien-28-oic acid (CDDO), CDDO-methyl (CDDO-Me), and CDDO-imidazolide (CDDO-Im) (Tanimoto indexes > 0.5) (Figure 5C), which are known activators of NRF2, key transcription factor responsible for cytoprotection and antioxidant defense response [36,37]. In accordance with this, further predictive modeling using the PASS online server suggested that Sol-DMAP potentially possesses hepatoprotective and cytoprotective activities, likely mediated through activation of the NRF2-dependent antioxidant response (Figure 5D).

To validate these predictions and elucidate the mechanism underlying the hepatoprotective activity of Sol-DMAP, we employed human hepatocellular carcinoma HepG2 cells as a hepatocyte-like cell model, as previously described [38]. Firstly, the cytotoxicity of Sol-DMAP in HepG2 cells was assessed to identify non-toxic concentrations for further experiments. The data showed that the investigated compound at concentrations up to 2 µM did not affect cell viability (Figure 5E). Treatment of HepG2 cells with Sol-DMAP markedly attenuated menadione (MEN)-induced cytotoxicity in a dose-dependent manner, restoring cell viability at 1 µM to near-control level (Figure 5F). Given that MEN-stimulated cytotoxicity is primarily mediated by oxidative stress [39], these results suggest that Sol-DMAP possesses antioxidant potential. Indeed, further gene expression analysis confirmed that Sol-DMAP at 0.5 µM significantly activated antioxidant genes in HepG2 cells, namely *HMOX1*, *GCLC*, *GCLM*, and *NQO1* by 2.3-, 2.9-, 5.2-, and 2.2-fold, respectively, compared to untreated control (Figure 5G). DOX also activated *NQO1* expression, and this induction was more potent than that elicited by Sol-DMAP alone (Figure 5G,H). Notably, in the combination treatment, Sol-DMAP significantly reduced the DOX-induced overexpression, returning *NQO1* to the level observed in Sol-DMAP-only group (Figure 5H). We hypothesize that the robust *NQO1* induction by DOX represents a compensatory antioxidant response in HepG2 cells. This response appears less pronounced in the presence of Sol-DMAP, likely because the triterpenoid effectively mitigates DOX-induced oxidative stress. Thus, our data demonstrate that Sol-DMAP not only activates the basal expression of antioxidant genes in hepatocyte-like cells but also attenuates DOX-induced activation of *NQO1*.

Based on (i) the obtained gene expression data (Figure 5G), (ii) the predicted NRF2-stimulating potency of Sol-DMAP (Figure 5D) and (iii) the reported ability of cyanoenone-bearing triterpenoids to directly interact with KEAP1, a constitutive repressor of NRF2 [40], we hypothesized that the hepatoprotective activity of Sol-DMAP is mediated by its disruptive effect on the KEAP1-NRF2 interaction. Molecular docking analysis confirmed that Sol-DMAP can bind to KEAP1 Kelch domain with low Gibbs free energy comparable to that of known KEAP1-NRF2 interaction inhibitor ML334 (−9.1 and −9.3 kcal/mol, respectively) (Figure 5I). Furthermore, Sol-DMAP formed hydrogen bonds with key amino acid residues crucial for KEAP1-NRF2 binding, namely Asn382, Arg415 and Arg483 (Figure 5J) [41,42]. In accordance with this, further protein-protein docking simulation independently confirmed that Sol-DMAP can affect KEAP1-NRF2 interaction. As shown in Figure 5K, Sol-DMAP prevents the formation of the conformationally correct complex by occupying NRF2-binding site in KEAP1 structure. In addition, Sol-DMAP decreased the thermodynamic stability of the KEAP1-NRF2 complex, as indicated by reduced Kd values and increased ΔG (Figure 5L). It is important to emphasize that the KEAP1-targeting potency of Sol-DMAP demonstrated here was predicted in silico and requires further experimental verification, including exploration of its effect on complete NRF2 signaling pathway.

Collectively, these results indicate that Sol-DMAP exhibits pronounced hepatoprotective potential, in addition to its P-gp-inhibiting activity and direct antitumor effect. This protective effect is closely associated with antioxidant properties of Sol-DMAP, which are hypothetically mediated through its inhibitory effect on the KEAP1-NRF2 interaction.

## 3. Discussion

MDR remains a critical obstacle to effective cancer chemotherapy, necessitating the development of novel therapeutic strategies [1]. Conventional inhibitors of P-gp, a major mediator of MDR, have been explored as potential agents to overcome resistance. However, their clinical success have been limited by insufficient efficacy and adverse side effects [10], highlighting the urgent need for new classes of P-gp modulators that are both potent and biocompatible. Natural compounds, especially those of triterpenoid origin, are of significant interest in this context due to their structural diversity, multitarget effects, and favorable safety profiles [43,44,45,46]. Their ability to regulate multiple tumor-associated signaling pathways makes them attractive scaffolds for developing next-generation MDR modulators that can restore chemosensitivity while ensuring systemic safety. Of special interest are P-gp inhibitors that, alongside their P-gp-targeting properties, can also exert a protective effect on P-gp-overexpressing organs, such as liver, kidneys, intestine, and adrenal glands.

In this study, we investigated the MDR-reversal activity of Sol-DMAP in RLS40 lymphosarcoma model in vivo. Our results clearly demonstrated that Sol-DMAP potentiated the antitumor efficacy of DOX, leading to a significant increase in the intratumoral DOX concentration, with the most pronounced effect observed on day 8 post-tumor implantation (Figure 2C). Surprisingly, this potentiating effect was significantly attenuated toward the end of the experiment (Figure 2C), which may be explained by either factors associated with progressive tumor growth or the revealed antioxidant potential of Sol-DMAP.

During active tumor growth, the ratio of its surface area to volume significantly decreases, which severely compromises the efficacy of chemotherapy [47]. Since P-gp is primarily expressed in perivascular tumor regions [48], its inhibition becomes less effective in inner, less vascularized parts of the tumor. In agreement with this, Patel et al. demonstrated that P-gp inhibitors mainly enhance DOX uptake in cells proximal to blood vessels but have little effect on distal tumor cells [49]. Given the marked acceleration of tumor growth in vehicle-administered mice toward the end of the experiment (Figure 2C; 11 and 13 days), inadequate penetration of Sol-DMAP into the deep layers of the tumor tissue could have caused the attenuation of its DOX-potentiating activity.

As noted previously, the antitumor effect of DOX is mediated, in part, by its ability to induce oxidative stress in tumor cells [50]. As shown in Figure 4F, Sol-DMAP effectively abrogated ROS production induced by DOX in RLS40 cells. This antioxidant effect may also underlie the reduced efficacy of the combined regimen during the later stages of the in vivo experiment. While Sol-DMAP initially enhances DOX accumulation via P-gp inhibition, its subsequent antioxidant activity, putatively associated with activation of the NRF2 signaling axis, may partially protect tumor cells from oxidative damage. Consistent with this, several studies have reported that NRF2 activation can induce DOX resistance in tumor cells [51,52,53]. Thus, the observed dual interplay requires further investigation to optimize a treatment regimen that effectively balances chemosensitization with redox modulation.

An important finding of this study is that Sol-DMAP not only potentiates the effect of DOX in vivo but also exerts its own comprehensive antitumor effect on RLS40 lymphosarcoma (Figure 2C). The demonstrated capacity of the triterpenoid to reduce both tumor volume (Figure 2C) and the mitotic activity of tumor cells (Figure 3C) in vivo is consistent with the in vitro data, which demonstrate the pro-apoptotic properties of Sol-DMAP (Figure 4B,C) and its ability to induce G1-phase cell cycle arrest (Figure 4G,H). These findings align with earlier reports describing the pro-apoptotic and cell cycle–inhibitory effects of structurally related triterpenoid analogs [32,54,55,56,57]. This combined action on both the MDR phenotype and tumor cell proliferation holds considerable promise for developing novel P-gp inhibitors with enhanced antitumor efficacy.

The second key aim of this study was to determine whether Sol-DMAP exerts a protective effect against DOX-induced liver damage. This research direction was motivated by several factors, including the expression of P-gp in liver tissue, the known liver-targeting properties of GA and its derivatives [23], and the critical importance of maintaining hepatic homeostasis during chemotherapy [58]. DOX and other chemotherapy drugs induce hepatotoxicity primarily through oxidative stress mediated by ROS activation [15,16,17], and disrupting efflux transporters like P-gp can potentially exacerbate this damage. P-gp is essential for normal liver function, promoting xenobiotic protection, biliary excretion, and drug metabolism [59,60]. Thus, broad inhibition of P-gp may enhance systemic exposure to P-gp substrates, increasing toxicity and, therefore, restricting the use of P-gp inhibitors [61]. Both clinical and experimental studies have demonstrated that combining P-gp inhibitors with DOX or other drugs can worsen liver injury [10,21,22], underscoring the necessity of a thorough safety assessing of the liver when developing new MDR modulators.

Our results demonstrated that Sol-DMAP provided valuable hepatoprotective effects. In DOX-treated mice, Sol-DMAP stimulated the hepatic antioxidant defense, as evidenced by increased Hmox-1 expression (Figure 5A, right panel) and reduced liver damage (Figure 5A, left panel; Table 1). In vitro studies confirmed that Sol-DMAP protects hepatocyte-like cells against menadione-induced oxidative toxicity (Figure 5F) and activates NRF2-depended antioxidant genes, namely *HMOX1*, *GCLC*, *GCLM*, and *NQO1* (Figure 5G). Molecular docking analysis further supported these findings by revealing that Sol-DMAP can bind to the KEAP1 Kelch domain, thereby disrupting the KEAP1-NRF2 complex (Figure 5I–K). Notably, this study evaluated the protective effect of Sol-DMAP only on the liver tissue of DOX-treated mice. A more comprehensive understanding of its protective potential requires further investigation into its effects on other organs with high P-gp expression. Our results, together with previously reported ability of CDDO-Im, a structural analog of Sol-DMAP, to induce NRF2 signaling in the murine kidney [62] and intestine [63], suggest a promising strategy for developing P-gp inhibitors based on cyanoenone-bearing triterpenoids with mitigated side effects on P-gp-overexpressing organs.

The present study has several limitations, which should be considered as future research perspectives. First, the KEAP1-targeting effect of Sol-DMAP was demonstrated here only using in silico approaches, which requires detailed experimental verification in subsequent studies. Second, a more detailed investigation of DOX accumulation in the presence of Sol-DMAP is needed. This should analyze different depths within tumor nodes and various time points, which could help in developing approaches to extend the duration of the DOX-potentiating effect of Sol-DMAP in vivo. Third, the antitumor potency of Sol-DMAP was explored here only in combination with DOX. Further experiments with other chemotherapeutic agents, particularly those for which ROS induction plays a less significant role in tumor growth suppression, are of great interest.

In conclusion, our study clearly demonstrates that P-gp inhibitor soloxolone *N*-3-(dimethylamino)propylamide (Sol-DMAP) exhibits MDR-reversing and hepatoprotective properties in vivo. Sol-DMAP effectively sensitized P-gp-overexpressing RLS40 lymphosarcoma to DOX by increasing its intratumoral accumulation without exacerbating systemic toxicity. Independently, Sol-DMAP displayed direct antitumor activity, inducing apoptosis and G1-phase cell-cycle arrest in RLS40 cells in vitro. Importantly, Sol-DMAP mitigated DOX-induced hepatic injury in RLS40-bearing mice. Mechanistic studies in HepG2 cells suggest that Sol-DMAP hypothetically modulates the NRF2-dependent antioxidant response, which is supported by its predicted ability to disrupt KEAP1-NRF2 interaction and the subsequent upregulation of NRF2-responsive genes. Collectively, these effects establish Sol-DMAP as a promising multifactorial agent for use as a biocompatible chemotherapy adjuvant.

## 4. Materials and Methods

### 4.1. In Vivo Experiments

#### 4.1.1. Mice

10–12 week-old CBA female mice were obtained from the vivarium of the Institute of Chemical Biology and Fundamental Medicine SB RAS (Novosibirsk, Russia). Mice were housed in plastic cages in standard daylight conditions (12/12 h light/dark cycle). Water and food were supplied ad libitum. All animal procedures were conducted in strict compliance with the guidelines for the proper use and care of laboratory animals (ECC Directive 2010/63/EU) and ARRIVE guidelines 2.0 [64,65]. The experimental protocols were approved by the Committee on the Ethics of Animal Experiments with the Institute of Cytology and Genetics SB RAS (ethical approval number 188 from 3 October 2024).

#### 4.1.2. Tumor Transplantation and Design of Animal Experiment

Tumors were induced in CBA mice (*n* = 10–12) by intramuscular (i.m.) injection of tumor cells (10^6^ cells/mouse) in 0.1 mL of PBS into the right thighs. On day 4 after tumor transplantation, the mice were divided into five groups (10–12 animals per group): (1) untreated control mice; (2) mice received intraperitoneal (i.p.) injections of 10% Tween-80 (vehicle); (3) mice received i.p. injections of Sol-DMAP in 10% Tween-80 at a dose of 20 mg/kg; (4) mice received i.p. injections of DOX in saline buffer at a dose of 5 mg/kg; (5) mice received combination therapy of Sol-DMAP and DOX at mentioned dosages with interval of 1 h. Treatment was administered three times a week; in total, five injections were performed. During the experiment, the tumor volumes were determined three times per week using caliper measurements and were calculated as V = (D × d^2^)/2, where D is the longest diameter of the tumor node and d is the shortest diameter of the tumor node perpendicular to D. Mice were sacrificed on day 14 after tumor transplantation and material (tumor nodes, livers, and kidneys) was collected for subsequent analysis.

#### 4.1.3. Toxicity Assessment

During the experiment, the general status and body weight of the animals were monitored. At the end of the experiment, livers and kidneys were collected, and organ indices were calculated as (organ weight/body weight) × 100%, followed by the normalization of organ indices of experimental mice to organ indices of control mice.

#### 4.1.4. Histology, Morphometry, and Immunohistochemistry

For the histological study, the tumor and liver specimens (*n* = 10 to 12 for each group) were fixed in 10% neutral-buffered formalin (BioVitrum, Moscow, Russia), dehydrated in ascending ethanols and xylols, and embedded in HISTOMIX paraffin (BioVitrum, Russia). Paraffin sections (up to 5 µm) were sliced on a Microm HM 355S microtome (Thermo Fisher Scientific, Waltham, MA, USA) and stained with hematoxylin and eosin.

For the immunohistochemical study, the tumor and liver sections (*n* = 3 for each group) were deparaffinized and rehydrated. Antigen retrieval was performed after exposure in a microwave oven at 700 W. The tumor samples were incubated with anti-P-gp primary antibodies (P7965, Sigma-Aldrich, St. Louis, MO, USA) according to the manufacturer’s protocol. The liver samples were incubated with anti-Hmox-1 primary antibodies (ab13248. Abcam, Cambridge, UK) according to the manufacturer’s protocol. Then, the sections were incubated with secondary horseradish peroxidase (HPR)-conjugated antibodies, exposed to the 3,30-diaminobenzidine (DAB) substrate (Rabbit Specific HRP/DAB (ABC) Detection IHC Kit, ab64261, Abcam, Boston, MA, USA), and stained with Mayer’s hematoxylin.

Morphometric analysis of tumor and liver sections was performed by point counting using a morphometric grid with 100 testing points in a testing area equal to 3.2 × 10^6^ μm^2^. Morphometric analysis of tumor tissue included the assessment of the volume densities (Vv, %) of necrosis and numerical densities (Nv) of mitosis. Morphometric analysis of liver tissue included the evaluation of the volume densities (Vv, %) of the unchanged liver tissue, dystrophy, necrosis, and blood vessels.

The volume density (Vv, %) of the histological structure studied indicates that the volume fraction of tissue occupied by this compartment is determined from the testing points lying over this structure and is calculated using the following formula, Vv = (P_structure_/P_test_) × 100%, where P_structure_ denotes the number of points over the structure and P_test_ denotes the total number of test points, 100 in this case. The numerical density (Nv) of the histological structure studied indicates the number of particles in the unit of tissue volume and is evaluated as the number of particles in the square unit, 3.2 × 10^6^ μm^2^ in this case. Ten random fields were examined from each liver or tumor specimen, forming 100 to 120 random fields for each group of mice.

The intensity of P-gp expression in the tumor tissue was assessed semiquantitatively using the following scale: 0—none; 1—mild; 2—moderate; 3—severe; and 4—total. A total of 10 random fields were examined from each tumor specimen, forming 30 random fields for each group of mice.

All histological images were examined and scanned using an Axiostar Plus microscope equipped with an AxioCam MRc5 digital camera (Zeiss, Oberkochen, Germany) at magnifications ×200 and ×400.

### 4.2. Identification of Compound Concentrations in Murine Blood and Tissues Using HPLS-MS/MS

#### 4.2.1. Blood Pharmacokinetics of Sol-DMAP

Mice were injected with Sol-DMAP at 20 mg/kg intraperitoneally. A 10 µL aliquot of blood was taken from the tail vein 15, 30 min, 1, 2, 3, 4, 6, and 24 h after Sol-DMAP administration and then processed according to the sample preparation protocol described below and analyzed.

#### 4.2.2. Preparation of Calibrators and Experimental Samples for Quantification of Sol-DMAP in Mice Blood

Sol-DMAP at 10 mM in DMSO was subjected to serial dilutions with methanol (Chimmed, Moscow, Russia) to give Sol-DMAP working solutions with concentrations of 50, 100, 200, 500, 1000, 2000, 2500, 5000 ng/mL and 10, 20, 50, and 100 µg/mL. Calibrators and quality control samples were prepared by mixing 90 µL of mice blood from a healthy control (blank) stabilized with EDTA with 10 µL of the corresponding working solution of the compound. The concentrations of the compound in calibrators were 5, 10, 20, 50, 100, 200, 500, 1000, 2000, and 5000 ng/mL and 10 µg/mL.

A measure of 10 μL of blood samples from control and experimental mice was applied to the WhatmanTM Protein Saver Card 903 paper carrier. The spots were dried at 25 °C for 3 h. Dried blood spots were completely excised and placed in 2 mL tubes, and 50 μL of water was added. The samples were shaken for 15 min, then 100 μL of methanol containing 0.1% formic acid (Panreac, Barcelona, Spain) and an internal standard (IS) (2-Ad (Sigma-Aldrich, St. Louis, MO, USA), 1000 ng/mL) (Figure 6) was added. The samples were shaken again for 15 min and centrifuged for 10 min at 13,400 rpm (MiniSpin, Eppendorf, Hamburg, Germany). An aliquot of 40 μL was transferred to a chromatographic vial for analysis.

#### 4.2.3. Preparation of Stock Solutions of Sol-DMAP and DOX and Internal Standard Working Solutions for HPLC-MS/MS Analysis of Tumors

The stock solution of Sol-DMAP was prepared by dissolving 1.0–1.1 mg (exact weight recorded) of the substance in a corresponding amount of 100% methanol to obtain a solution with 1.0 mg/mL of the substance. DOX (Sigma-Aldrich, St. Louis, MO, USA) in physiological saline (5 mg/mL) was diluted with methanol to obtain 1.0 mg/mL.

For the internal standard (IS) working solution, an exact amount of 2-adamantilamine hydrochloride (2-Ad, Figure 6) was dissolved in methanol and serially diluted for final concentrations of 1000 ng/mL (for blood sample preparation) and 20 µg/mL (for tissue homogenate preparation). All solutions were stored at −18 °C and equilibrated to room temperature prior to sample preparation.

#### 4.2.4. Tumor Homogenization Protocol

A sample of mouse tumor tissue was placed in a polypropylene tube, and 400 µL of physiological saline was added per 100 mg of tissue. The sample was homogenized using an Alyona ultrasonic homogenizer (Center of Ultrasonic Technologies LLC, Biysk, Russia). Homogenization was carried out at 100% power (c.a. 60 W) with three 5 s pulses (duration of 5 s), with cooling intervals to prevent overheating.

#### 4.2.5. Preparation of Calibrators and Experimental Samples for Quantification of Sol-DMAP and DOX in Tumors

Aliquots of equal volume of Sol-DMAP and DOX stock solutions were mixed, and working solutions containing both compounds at concentrations of 1, 10, 50, 100, 500, and 1000 ng/mL were prepared by subsequent dilutions with methanol. Calibrators were prepared by adding 10 µL of the corresponding working solution to 100 µL of blank homogenate, resulting in final concentrations of 1, 10, 50, 100, 500, and 1000 ng/g of each compound. Samples obtained from the experimental animals were prepared as described below.

To a 100 µL aliquot of tissue homogenate, 30 µL of internal standard solution (2-Ad, 20 µg/mL) was added, and the samples were mixed and incubated for 10 min at ambient temperature. Subsequently, 400 µL of acetonitrile (containing 0.1% formic acid (Chimmed, Moscow, Russia)) and 100 mg of QuEChERS salts (Interlab, Moscow, Russia) were added. The mixture was mixed again and shaken for 30 min. After centrifugation (10 min, 13,400 rpm [12,044 g], MiniSpin, Eppendorf, Hamburg, Germany), 300 µL of the supernatant was transferred to a 1.5 mL polypropylene tube containing 50 mg of QuEChERS sorbent. The mixture was shaken for 10 min and centrifuged under the same conditions. Finally, 20 µL of the resulting supernatant was diluted with 180 µL of methanol in a vial insert. An injection volume of 10 µL was used for analysis.

#### 4.2.6. Instrumentation and LC-MS/MS Conditions

The chromatographic system used in the study consisted of a Shimadzu LC-20AD Prominence chromatograph equipped with a cooled autosampler, a binary gradient pump, and a column oven thermostated at 40 °C. Detection was performed using a SCIEX 6500 QTRAP mass spectrometer (SCIEX, Framingham, MA, USA). Chromatographic separations of Sol-DMAP and the internal standard (2-Ad) were achieved on a ProntoSil 120-5 C18 AQ column (2 × 75 mm, 5 µm, EcoNova, Novosibirsk, Russia). The mobile phase consisted of (A) water with 0.1% formic acid (*v*/*v*) and (B) methanol with 0.1% formic acid (*v*/*v*). The following gradient was applied: 0 min—3% B; 3.5 min—95% B; 7 min—95% B. The flow rate and injection volume were 250 µL/min and 10 µL, respectively. After a chromatographic analysis, the column was equilibrated for an additional 3 min step.

Data were acquired in multiple reaction monitoring (MRM) mode with positive electrospray ionization. The mass spectrometer settings were as follows: curtain gas (CUR) = 30 psi, collision-activated dissociation gas (CAD) = high, ion source voltage (IS) = 5500 V, temperature (TEM) = 300 °C, sprayer gas (GS1) = 20 psi, evaporator gas (GS2) = 20 psi. The parameters for the detection of Sol-DMAP, DOX, and 2-Ad are given in Appendix A. Instrument control and data acquisition were managed using Analyst MD 1.6.3 software (AB SCIEX), and data processing was performed with MultiQuant 2.1 software (AB SCIEX).

### 4.3. In Vitro Experiments

#### 4.3.1. Cell Cultures and Evaluated Compound

Murine lymphosarcoma cells with MDR phenotype RLS40 cells were obtained from the cell bank of the Institute of Chemical Biology and Fundamental Medicine SB RAS (Novosibirsk, Russia). Human hepatocellular carcinoma HepG2 cells, human embryonic kidney HEK293 cells, Madin-Darby canine kidney MDCK cells, and murine J774 macrophage cell line were obtained from the Russian Culture Collection (Institute of Cytology RAS, St. Petersburg, Russia). The cells were cultured in Dulbecco’s modified Eagle’s medium (DMEM) (Sigma Aldrich, St. Louis, MI, USA) (HepG2, HEK293, MDCK, J774) and Iscove’s Modified Dulbecco’s Medium (IMDM) (Sigma Aldrich, St. Louis, MO, USA) (RLS40) containing 10% (*v*/*v*) heat-inactivated fetal bovine serum (Dia-M, Moscow, Russia), 10,000 IU/mL penicillin, 10,000 μg/mL streptomycin, and 25 μg/mL amphotericin (MP Biomedicals, Illkirch-Graffenstaden, France). For the passing of RLS40 cells, the culture medium was supplemented with 40 nM vinblastine. The cells were incubated at 37 °C in 5% CO_2_. GA derivative Sol-DMAP was synthesized according to the published protocol [32] with a purity ≥ 98% (HPLC UV). Sol-DMAP was dissolved in DMSO at 10 mM (stock solution) and kept at −20 °C until use.

#### 4.3.2. Cell Viability Assay

The cells were seeded in 96-well plates at 10^4^ cells/well in IMDM (RLS40) or DMEM (HepG2, HEK293, MDCK, J774) with 10% fetal bovine serum for 24 h. Then, the medium was replaced with serum-free medium to eliminate proliferation stimuli, and the cells were treated for an additional 24 h with either DOX (10–80 μM) (Sigma-Aldrich, St. Louis, MO, USA), Sol-DMAP (0.5–80 μM), their combination, or a combination with *N*-acetyl-L-cysteine (NAC) (2 mM). After the treatment period, 3-(4,5-dimethylthiazol-2-yl)-2,5-diphenyltetrazolium bromide (MTT) was added to a final concentration of 0.5 mg/mL for 2 h. Finally, the formazan crystals formed by MTT reduction in live (metabolically active) cells were dissolved in DMSO, and the optical density was measured at 570 and 620 nm using a Multiscan RC plate reader (Thermo LabSystems, Helsinki, Finland).

To access the antioxidant effect, HepG2 cells were preincubated with Sol-DMAP (0.25–1 μM) for 24 h, then the medium was replaced with fresh medium containing oxidative stress inducer menadione (MEN, 60 μM) for an additional 6 h, followed by the measurement of cell viability using the MTT test as described above.

#### 4.3.3. Apoptosis Assay

RLS40 cells were seeded in 24-well plates at 10^5^ cells/well in IMDM with 10% fetal bovine serum. After 24 h incubation, the cells were treated with DOX and Sol-DMAP at concentrations close to their twofold IC_50_ values (250 and 80 µM, respectively), or with a combination of both, followed by incubation for an additional 24 h. Thereafter, the cells were harvested by trypsinization (TrypLE Express (Gibco, Grand Island, NY, USA)) and centrifugation at 400× *g* for 7 min, washed with PBS, and stained with annexin V and propidium iodide (PI) using Annexin V-FITC Apoptosis Staining/Detection Kit (Abcam, Cambridge, UK) according to the manufacturer’s instructions. In brief, cells were resuspended in 500 μL of 1× binding buffer and incubated with 3 μL of annexin V-FITC and 5 μL of PI for 15 min at room temperature in the dark. The samples were analyzed using a NovoCyte Flow Cytometer (ACEA Biosciences Inc., San Diego, CA, USA), with at least 10,000 events for each sample.

#### 4.3.4. Caspase-3/-7 Activity Assay

The activity of caspase-3/-7 was evaluated using the Caspase-Glo 3/7 assay kit (Promega, Madison, WI, USA) according to the manufacturer’s instructions. Briefly, RLS40 cells were seeded in a black-walled 96-well plate at 10^4^ cells/well, incubated for 24 h, and then treated with DOX (250 μM), Sol-DMAP (80 μM), or their combination for 6 h. Thereafter, 100 µL of Caspase-Glo 3/7 Reagent was added to each well with subsequent incubation for 30 min in the dark. The luminescence was measured using a CLARIOstar plate reader (BMG Labtech, Ortenberg, Germany).

#### 4.3.5. ROS Accumulation Assay

Intracellular ROS levels were measured using the DCFDA/H2DCFDA Cellular ROS Assay Kit (Abcam, Cambridge, UK). RLS40 cells were incubated with Sol-DMAP (80 μM) for 24 h. Thereafter, cells were washed with PBS and incubated with 20 µM of H_2_DCFDA for 30 min under standard conditions. Then, the medium was removed and cells were washed with PBS and analyzed using the NovoCyte Flow Cytometer, with at least 10,000 events for each sample.

#### 4.3.6. Cell Cycle Analysis

RLS40 cells were seeded in 24-well plates at 10^5^ cells/well in IMDM with 10% fetal bovine serum. After 24 h incubation, cells were treated with Sol-DMAP (80 μM) for an additional 24 h. Thereafter, cells were detached with TrypLE Express and fixed in cold 70% ethanol overnight. The cells were resuspended in PBS, and 4′,6-diamidino-2-phenylindole (DAPI, Thermo Fisher Scientific, Dreieich, Germany) was added at 10 µg/mL for 2 h. Samples were analyzed using the NovoCyte Flow Cytometer, with at least 10,000 events for each sample.

#### 4.3.7. Total RNA Isolation

Total RNA was isolated from HepG2 cells after incubation with Sol-DMAP for 24 h using TRIzol Reagent (Ambion, Austin, TX, USA) according to the manufacturer’s protocol. In the case of RLS40 tumors of experimental animals, tumor tissues were collected in 1.5 mL tubes, filled with 1 g/tube of lysing matrix D (MP Biomedicals, Irvine, CA, USA) and 1 mL/tube of TRIzol reagent and homogenized using a FastPrep-24 TM 5G homogenizer (MP Biomedicals, Irvine, CA, USA). Thereafter, the homogenates were transferred to the new 1.5 mL tubes without lysing matrix and total RNA extraction was performed using TRIzol reagent according to the manufacturer’s protocol.

#### 4.3.8. Quantitative Real-Time PCR (RT-qPCR)

Firstly, the cDNA strand was synthesized from 4 µg of total RNA template using the M-MuLV-RH reverse transcriptase (Biolabmix, Novosibirsk, Russia) and random hexa-primers. RT-qPCR was carried out using BioMaster SYBR Blue reagent kit (Biolabmix, Novosibirsk, Russia) and specific primers (Appendix A) for RLS40 and HepG2 cells according to the previously described protocol [66]. For tumor tissue, RT-qPCR was carried out using HS-qPCR (2×) master mix (Biolabmix, Novosibirsk, Russia) and forward and reverse primers to *Tbp* and *Tbp*-specific ROX-labeled probe, each of the forward and reverse gene-specific primers, and FAM-labeled probe (Appendix A) according to the previously described protocol [67]. Relative expression was determined by normalization to the housekeeping gene *HPRT1* or *Tbp* using the comparative cycle threshold method (2^−ΔΔCt^).

### 4.4. In Silico Prediction

#### 4.4.1. Molecular Docking

Protein–ligand docking: Crystal structure of KEAP1 (PDB ID: 4L7B) complexed with ML334 [(1S,2R)-2-[(1S)-1-[(1,3-dioxoisoindol-2-yl)methyl]-3,4-dihydro-1H-isoquinoline- 2-carbonyl]cyclohexane-1-carboxylic acid)] was retrieved from the RCSB PDB database (https://www.rcsb.org/, accessed on 24 April 2025). Co-crystallized water and ligands were removed using Discovery Studio Visualizer v. 17.2.0 following the addition of polar hydrogen atoms and Gasteiger charges using AutoDockTools v.1.5.7. Sol-DMAP and ML334 structures were created using Marvin Sketch 5.12, and their geometry was optimized using Avogadro 1.2.0 (MMFF94 force field). All rotatable bonds were allowed to be freely flexible using AutoDockTools. The grid box of 18 × 18 × 18, centered at [x, y, z = −3.561030, 2.506091, 27.500515], was set to cover the ML334 binding pocket. Molecular docking of Sol-DMAP and ML334 was performed using AutoDock Vina [68]. The docking results were analyzed and visualized using the Discovery Studio Visualizer 17.2.0.

Protein–protein docking: ZDOCK web server [69] (https://zdock.wenglab.org/, accessed on 12 May 2025) was used to perform molecular docking simulations for predicting interactions between the KEAP1 Kelch domain and the NRF2 N-terminal domain, obtained from the RCSB PDB database (PDB ID: 2FLU, accessed on 26 April 2025). Firstly, NRF2 and KEAP1 were separated using Discovery Studio Visualizer v.17.2.0 and the Sol-DMAP molecule was added to KEAP1 according to protein–ligand docking results. Protein–protein docking was performed using the NRF2 N-terminal domain and KEAP1 with or without Sol-DMAP using ZDOCK 3.0.2 version. The docking results were analyzed and visualized using the Discovery Studio Visualizer 17.2.0.

#### 4.4.2. Structure Similarity Analysis

The structural similarity of Sol-DMAP with known NRF2 activators was evaluated by the Tanimoto score calculation using ChemBioServer 2.0 (https://chembioserver.vi-seem.eu/, accessed on 13 May 2025) [70].

#### 4.4.3. Biological Activity Prediction

Possible biological activities of compounds were evaluated using the PASS online tool (https://way2drug.com/PassOnline/, accessed date: 15 May 2025) [71]. Values of Pa (Probable activity) > Pi (probable inactivity) and Pa > 0.5 were considered to indicate biological activity for a compound.

### 4.5. Statistical Analysis

Statistical analysis was performed using GraphPad Prism version 8.0.1 (GraphPad Software, San Diego, CA, USA). The normality of data was assessed with the Shapiro–Wilk test. The significance of differences between the groups with normal and non-normal distribution was analyzed using unpaired Student’s *t*-test and non-parametric Mann–Whitney U-test, respectively. The differences were considered significant at *p* < 0.05.

## Figures and Tables

**Figure 1 ijms-26-11912-f001:**
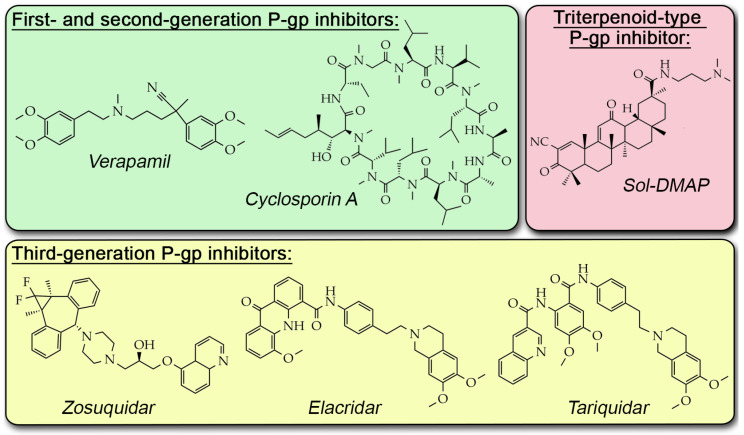
Chemical structures of known P-gp inhibitors.

**Figure 2 ijms-26-11912-f002:**
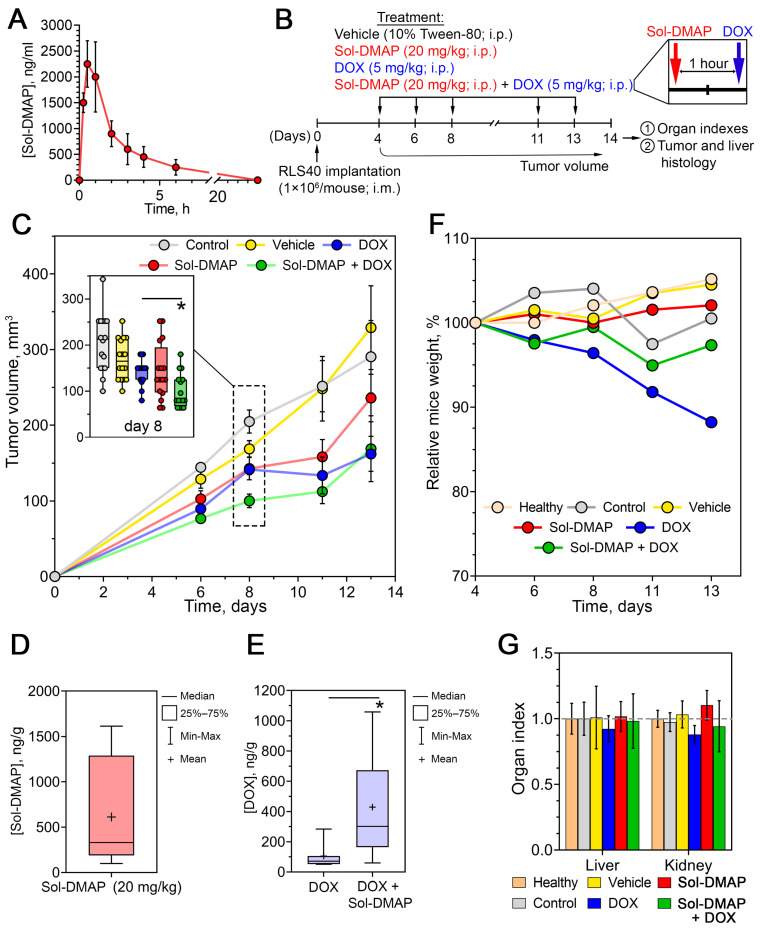
Antitumor effect of Sol-DMAP and its combination with DOX in murine RLS40 lymphosarcoma in vivo. (**A**) Pharmacokinetics of Sol-DMAP in blood of experimental animals following its intraperitoneal (i.p.) administration (20 mg/kg). (**B**) Experimental setup. (**C**) Dynamics of RLS40 tumor volumes in control and experimental groups. The inset box plot in the upper left corner shows tumor volumes on day 8. (**D**) The level of Sol-DMAP in tumor tissue measured 1 h after i.p. administration (20 mg/kg). (**E**) The level of doxorubicin (DOX) in mouse tumor tissue measured 1 h after administration with or without Sol-DMAP. (**F**,**G**) Body weights (**F**) and organ indexes (**G**) of RLS40 lymphosarcoma bearing mice without treatment and after administration of Sol-DMAP, DOX, or their combination; * *p* < 0.05.

**Figure 3 ijms-26-11912-f003:**
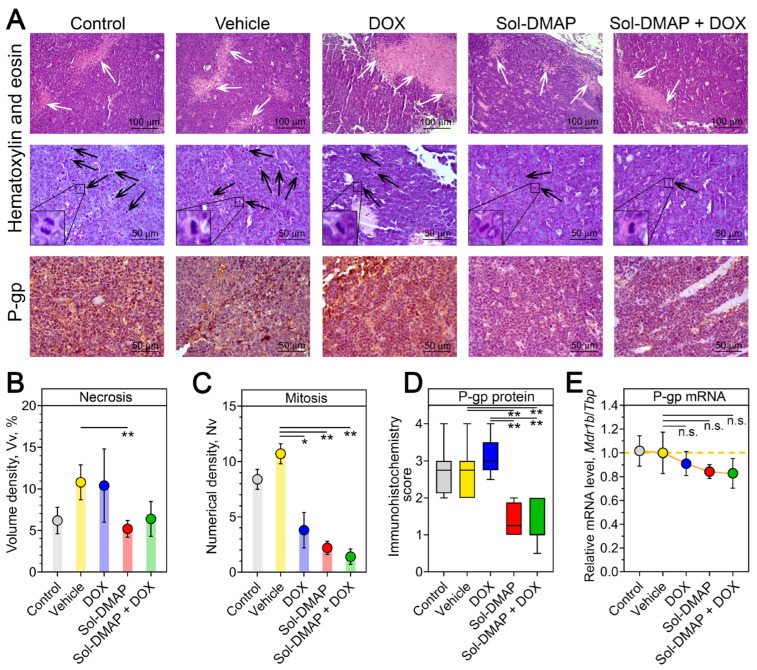
Structural alterations in the tumor tissue of RLS40-bearing mice receiving Sol-DMAP or Sol-DMAP+DOX. (**A**) Representative images of RLS40 tumors in control and experimental animals. Hematoxylin and eosin staining (upper and middle panel) and immunohistochemical staining with primary antibodies to P-gp (bottom panels). Original magnification ×200 (upper panel) and ×400 (middle and bottom panels). White and black arrows indicate the areas of necrosis in the tumor tissue and tumor cells in the state of mitosis, respectively. Representative mitoses at higher magnification are shown in lower left corner. (**B**,**C**) The volume density ((**B**), Vv, %) of necrosis and the numerical density ((**C**), Nv) of mitoses in the tumor tissue of control and experimental animals. (**D**) The intensity of P-glycoprotein (P-gp) expression in the tumor tissue of control and experimental animals assessed semiquantitatively. (**E**) Relative mRNA level of *Mdr1b*/*Tbp* in tumor tissue of control and experimental animals evaluated by RT-qPCR. * *p* < 0.05; ** *p* < 0.01. n.s.—not significant.

**Figure 4 ijms-26-11912-f004:**
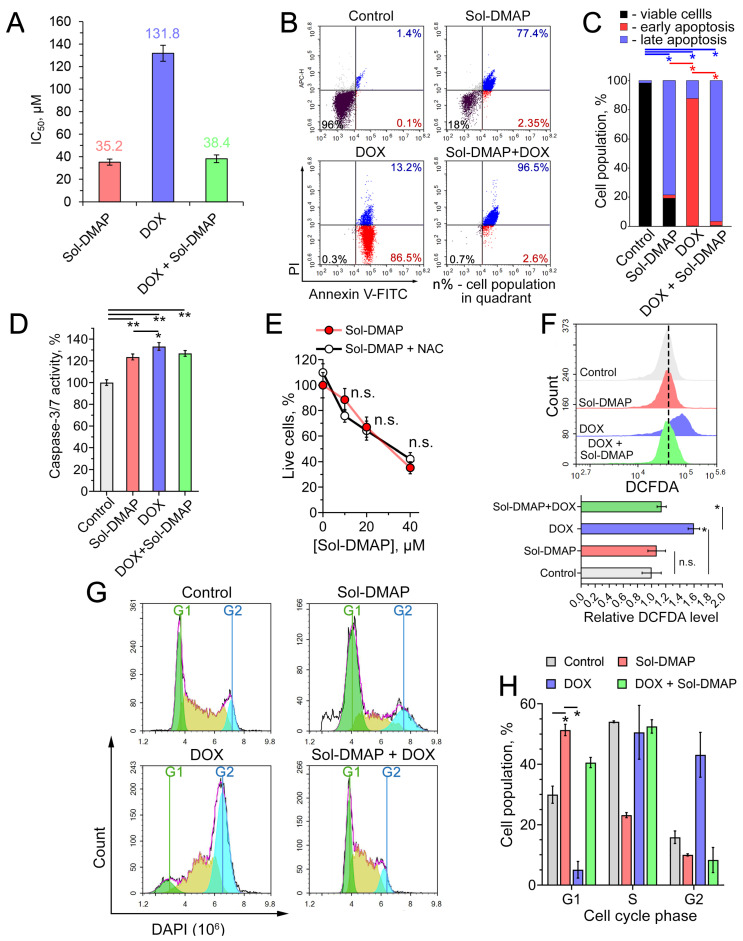
Evaluation of antitumor effect of Sol-DMAP in RLS40 cells in vitro. (**A**) IC_50_ values of Sol-DMAP, DOX, and their combination (DOX + 20 µM Sol-DMAP) in RLS40 cells (incubation for 24 h). (**B**,**C**) Effect of Sol-DMAP (80 µM), DOX (250 µM), and their combination on the apoptosis induction in RLS40 cells assessed by flow cytometry analysis. Annexin V-FITC and propidium iodide double staining. Representative cytograms (**B**) and percentage of cell distribution between viable, early apoptotic, and late apoptotic cells (**C**). Red- and blue-colored pairwise comparisons represent statistical differences in early and late apoptosis stages, respectively. (**D**) Activation of caspase-3/-7 in RLS40 cells incubated with Sol-DMAP (80 µM), DOX (250 μM), and their combination for 6 h. (**E**) The impact of reactive oxygen species (ROS) inhibition by N-acetyl-L-cysteine (NAC) (2 mM) on the 24 h cytotoxicity of Sol-DMAP in RLS40 cells. (**F**) ROS production in RLS40 cells after 24 h of incubation with Sol-DMAP (80 μM), DOX (250 µM), and their combination assessed by flow cytometry after DCFDA staining. Representative histogram (upper part) and relative DCFDA level (lower part). (**G**,**H**) Cell cycle phase analysis of RLS40 cells after incubation with Sol-DMAP (80 μM), DOX (250 µM), and their combination for 24 h. Representative histograms representing cell cycle distribution (**G**) and the percentage of cells in each phase of the cell cycle (**H**). * *p* < 0.05; ** *p* < 0.01. n.s.—not significant.

**Figure 5 ijms-26-11912-f005:**
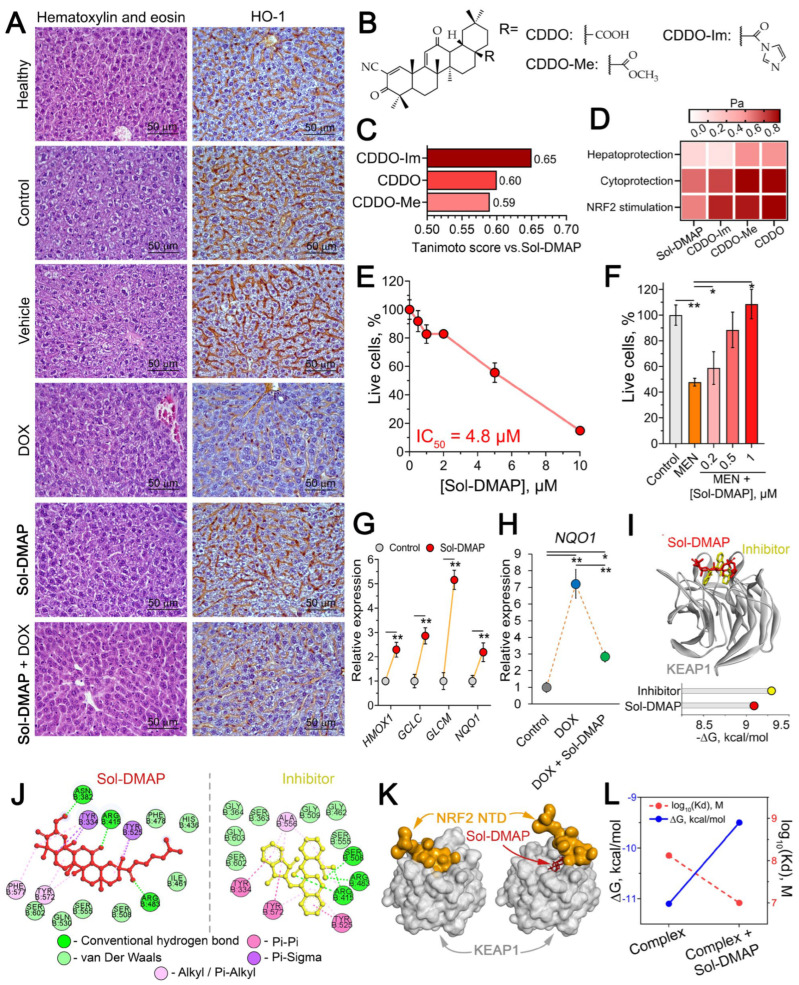
Hepatoprotective effect of Sol-DMAP in vivo and in vitro. (**A**) Morphological changes in the liver tissue of healthy and RLS40-bearing mice after Sol-DMAP administration. Hematoxylin and eosin staining (left panel) and immunohistochemical staining with primary antibodies to Hmox1 (right panel). Original magnification ×400. (**B**) Structures of known triterpenoid-type NRF2 activators. (**C**) Tanimoto similarity of Sol-DMAP with known triterpenoid-type NRF2 activators calculated using ChemBioServer 2.0. (**D**) Heatmap illustrating predicted biological activities of Sol-DMAP and its analogs using PASS online tool. Pa—probability of being active. (**E**) Cytotoxicity of Sol-DMAP in HepG2 cells evaluated by MTT assay (24 h). (**F**) Protective effect of Sol-DMAP on menadione (MEN)-induced cytotoxicity in HepG2 cells evaluated by MTT assay (24 h). (**G**) Expression levels of antioxidant genes in HepG2 cells after 24 h of incubation with 0.5 μM Sol-DMAP, as measured by RT-qPCR. *HPRT1* was used as a reference gene. (**H**) Relative expression of *NQO1* in HepG2 cells after 24 h of incubation with DOX (10 μM) and DOX + Sol-DMAP (0.5 μM) measured by RT-qPCR. (**I**) The binding mode of Sol-DMAP and ML334 (reference inhibitor) to the KEAP1 kelch domain and Gibbs free energy (ΔG) of their interaction. (**J**) Two-dimensional diagrams of the molecular interaction between Sol-DMAP or ML334 (reference inhibitor) to the KEAP1 kelch domain predicted by molecular docking. (**K**) Interaction of KEAP1 with NRF2 N-terminal domain in the presence or absence of Sol-DMAP in KEAP1 kelch domain. (**L**) Affinity (Kd) and ΔG of KEAP1-NRF2 complex in the presence or absence of Sol-DMAP in KEAP1 kelch domain. Protein-protein molecular docking was performed using ZDOCK Server web tool. * *p* < 0.05; ** *p* < 0.01.

**Figure 6 ijms-26-11912-f006:**
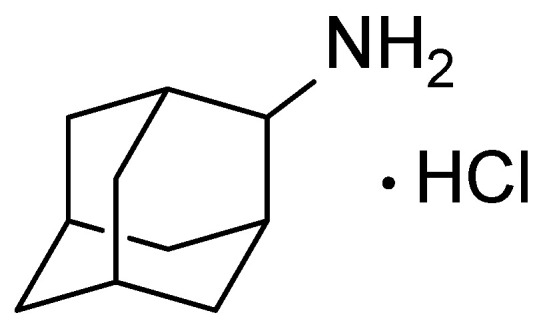
The structure of 2-adamantilamine hydrochloride (2-Ad, IS).

**Table 1 ijms-26-11912-t001:** Structural changes in the liver tissue of RLS40 lymphosarcoma-bearing mice without treatment and after administration of DOX, Sol-DMAP, or their combination.

	Healthy	Control	Vehicle	DOX	Sol-DMAP	Sol-DMAP + DOX
Unchanged liver tissue, Vv, %	71.2 ± 2.3	24.7 ± 1.6	33.9 ± 1.4	33.4 ± 3.3 *	42.2 ± 3.3 **#	56 ± 2.8 **##^^
Dystrophy, Vv, %	7 ± 0.6	37.9 ± 1.9	27.5 ± 1.9	23.1 ± 2.9 **	16.9 ± 2.3 **#	10.5 ± 1.3 **##^
Necrosis, Vv, %	7.6 ± 1.5	24.8 ± 1.2	26.8 ± 1.1	20.9 ± 1.7 #	24 ± 1.1	16 ± 3 *#
Total destructive changes, Vv, %	14.6 ± 1.5	62.7 ± 1.7	54.6 ± 1.7	44 ± 3.9 **#	40.9 ± 3.1 **##	26.5 ± 4.2 **##^
Blood vessels, Vv, %	4.3 ± 0.4	6.1 ± 0.6	5.3 ± 0.5	15.5 ± 0.7 **##	5.8 ± 0.6 ^^	5.8 ± 0.4 ^^
Other, Vv, %	9.9 ± 0.8	6.6 ± 0.3	6.6 ± 0.6	7 ± 0.8	11.1 ± 0.9 **##^	11.8 ± 1.7 *#^

Vv, %—the volume density representing the volume fraction of tissue occupied by the studied compartment. Statistically significant differences from control at *p* ≤ 0.05 (*), *p* ≤ 0.001 (**); from vehicle at *p* ≤ 0.05 (#), *p* ≤ 0.001 (##); from DOX at *p* ≤ 0.05 (^), *p* ≤ 0.001 (^^).

## Data Availability

The original contributions presented in this study are included in the article/Appendix A. Further inquiries can be directed to the corresponding author.

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
