# Peer review of "Soloxolone *N*-3-(Dimethylamino)propylamide Suppresses Tumor Growth and Mitigates Doxorubicin-Induced Hepatotoxicity in RLS40 Lymphosarcoma-Bearing Mice"

_ijms, 2025, doi:10.3390/ijms262411912_

Round 1

Reviewer 1 Report

Comments and Suggestions for Authors

This is a well structured article but I have major comments that should be addressed:

  • You didn’t provide a clear data on how NRF2 pathway effected in Sol-DMAP-treated RLS40 cells. Why you didn’t measured the complete pathway in vivo and in vitro to confirm your conclusion.
  • You didn’t mentioned if you used any reference antitumour drugs for the in vitro and in vivo work
  • Why you start your data  results by the in vivo, not in vitro work? While methods by in vitro.
  • In figure 2 C it is not clear the inserted figure in the description of figure legened.
  • Can you clearly status how you measured cell viablity and why you measured in cell free media
  • You mentioned that Sol-DMAP is well-tolerated in RLS40 lymphosarcoma-bearing mice with no effect on liver and kidney indices. Why did you specify this, and why didn't you mention its effect on hematological indices?
  • The text states that Sol-DMAP caused a slight decreasein P-gp that was not statistically significant (Fig 3A, D). The immunohistochemistry images in Fig 3A, however, appear to show a more substantial visual reduction in P-gp staining in the Sol-DMAP and others groups. The authors should illustrate this discrepancy between the qualitative images and the quantitative PCR data.
  • You used one hour time point for tumor analysis, could you clarify why?
  • Fig 4F legend: relative DCFDA level (lower partd) should be (lower part).
  • All the results need to be easily read with  informative legend because it contains many data that is not easy to be followed, and also clarify symbols for example (+) in Fig 2D.
  • Discussion needed to be rearranged as follows: tumour geometry Nrf2 hypotheses and finally future research and commendation.

Author Response

Dear Reviewer #1,

We are profoundly grateful for your thorough analysis of our manuscript and for identifying several aspects that required further development. Your valuable comments and suggestions have been instrumental in significantly strengthening the manuscript and improving its clarity for the reader.

All revisions made in response to your comments and those of the second reviewer have been highlighted in yellow within the text.

Please find our point-by-point responses to your feedback below.

You didn’t provide a clear data on how NRF2 pathway effected in Sol-DMAP-treated RLS40 cells. Why you didn’t measured the complete pathway in vivo and in vitro to confirm your conclusion.

Authors: Corrected. Indeed, the NRF2-targeting effect of Sol-DMAP was demonstrated in our work only indirectly, as a proposed hypothesis, using chemoinformatic approaches (Fig. 5C, D), molecular docking (Fig. 5I-L), and measurement of the expression of NRF2-dependent antioxidant genes (Fig. 5G, H) and the Hmox1 protein (Fig. 5A). Since the potential link between the hepatoprotective activity of Sol-DMAP and the KEAP1/NRF2 signaling axis was identified at a late stage of the research, we were unable to initiate a detailed experimental workstream to confirm this interaction via western blot and/or confocal microscopy. Such work is the objective of our subsequent studies.

In accordance with your comment, and to avoid misleading the readers, we have made adjustments throughout the manuscript text to emphasize the hypothetical nature of the results suggesting a probable KEAP1/NRF2-targeting effect of Sol-DMAP (please see p. 1, lines 31-32 (Abstract); p. 12, lines 388-391; p. 12, lines 395-396; p. 13, lines 433-434; p. 14, lines 496-499 (Conclusion)). Moreover, in the Discussion, we have added a paragraph dedicated to the limitations of the study, where the in silico nature of the data concerning the Sol-DMAP/KEAP1/NRF2 link is listed as the first point (p. 14, lines 480-482). We believe the revised text now makes it clearer to readers that while Sol-DMAP's effect on KEAP1/NRF2 is a plausible explanation for our observations (hepatoprotection) in the liver cells, this hypothesis requires experimental confirmation in future studies.

You didn’t mentioned if you used any reference antitumour drugs for the in vitro and in vivo work

Authors: Thank you for raising this important question. The primary objectives of our in vivo study were to determine (a) whether Sol-DMAP possesses intrinsic antitumor potential and (b) whether it can enhance the intratumoral accumulation and efficacy of DOX in RLS40 lymphosarcoma-bearing mice.

To address the first objective, we used DOX as a reference antitumor drug and demonstrated that Sol-DMAP has comparable, though slightly weaker, efficacy (Fig. 2C). DOX is a cornerstone of first-line treatment regimens for a wide range of hematological malignancies (e.g., CHOP, R-CHOP, AIDA, 7+3, ALL-2009), underscoring its relevance and well-established role as a reference drug. Regarding the effect of Sol-DMAP on the intratumoral accumulation of DOX (Fig. 2E), we decided not to include a reference P-glycoprotein (P-gp) inhibitor. This decision was based on our previous work, which revealed that Sol-DMAP inhibits P-gp pump activity with potency comparable to or slightly greater than the reference inhibitor verapamil [1].

Considering your comment, we agree that including an additional animal group treated with verapamil or another, more selective P-gp inhibitor would have further strengthened our findings. We thank you for this valuable observation. In our subsequent studies, we will adopt a more meticulous approach to the design of in vivo experiments.

Why you start your data results by the in vivo, not in vitro work? While methods by in vitro.

Authors: Corrected. We appreciate your thorough review of the manuscript. Indeed, the structure of the Materials and Methods section in the original version was organized in a classical format (Materials / Cells / Animals / In Silico), which did not align with the logical sequence of the experimental workflow. In accordance with your comment, we have now restructured the Materials and Methods section (please see pp. 15-19). The revised organization now follows the chronological and methodological flow of the experiments.

In figure 2 C it is not clear the inserted figure in the description of figure legened.

Authors: Corrected. The inset box plot in Fig. 2C shows a more detailed distribution of tumor volumes among control and experimental animals on day 8, when the DOX-potentiating effect of Sol-DMAP was observed. In accordance with your suggestion, and to make this clearer for the reader, we have expanded the description of this box plot in the figure legend (please see p. 4, line 128) and added the label "day 8" beneath the inset (Fig. 2C).

Can you clearly status how you measured cell viablity and why you measured in cell free media

Authors: Corrected. We acknowledge that the description of cell viability measurement in the original manuscript was overly brief, which could have led to confusion regarding the assay's underlying principles.

As suggested, we have added a more detailed explanation of the MTT assay methodology, specifically clarifying the formation of formazan crystals and their relationship to the number of metabolically active (viable) cells. Please see p. 18, lines 676-677.

Regarding the absence of FBS in the cell medium during incubation with Sol-DMAP, this approach was chosen to assess the cytotoxicity of the triterpenoid in the absence of proliferation stimuli such as serum. Furthermore, FBS deprivation synchronizes the cell population, allowing for a more accurate evaluation of the cytotoxic profile of small molecules. This experimental design is consistent with MTT assay protocols used in previous studies (e.g., [2,3]).

You mentioned that Sol-DMAP is well-tolerated in RLS40 lymphosarcoma-bearing mice with no effect on liver and kidney indices. Why did you specify this, and why didn't you mention its effect on hematological indices?

Authors: Given that the liver and kidneys are the principal organs for the metabolism, detoxification, and elimination of xenobiotics, including doxorubicin (DOX) [4–6], we selected the assessment of liver and kidney indices (Fig. 2G), followed by histological analysis of liver tissue (Fig. 5A), as key markers for evaluating the tolerability of the treatment regimens. This methodological approach was further informed by the well-documented hepatoprotective and nephroprotective properties of glycyrrhetinic acid and its derivatives [7], as well as other phytochemicals [8,9].

We acknowledge that measuring serum levels of ALT/AST or creatinine/urea in mice would provide a more detailed view of Sol-DMAP's tolerability profile. However, such an analysis was beyond the scope of the present study, and we currently lack the requisite biological material (blood or serum) to perform these ELISA assays. In light of this valuable comment, the exploration of hematological indices of Sol-DMAP tolerability will be incorporated into subsequent, more comprehensive studies focusing on the safety and pharmacokinetics of Sol-DMAP.

The text states that Sol-DMAP caused a slight decrease in P-gp that was not statistically significant (Fig 3A, D). The immunohistochemistry images in Fig 3A, however, appear to show a more substantial visual reduction in P-gp staining in the Sol-DMAP and others groups. The authors should illustrate this discrepancy between the qualitative images and the quantitative PCR data.

Authors: Corrected. Indeed, as you correctly noted, the immunohistochemical analysis revealed more pronounced differences in P-gp expression in the Sol-DMAP-treated groups (Fig. 3A) compared to the RT-qPCR data (Fig. 3E).

To investigate this discrepancy in greater detail, we performed an additional semiquantitative assessment of P-gp staining intensity in tumor sections using the following scale: 0 – none, 1 – mild, 2 – moderate, 3 – severe, and 4 – total. The results of this analysis allowed us to conclude that Sol-DMAP statistically significantly suppressed P-gp expression at the protein level in tumor tissue (Fig. 3D). The weaker response at the mRNA level (Fig. 3E) may be explained by the timing of tumor tissue collection at the final stage of the experiment. By this endpoint, the cellular response to chemotherapy at the protein level was well-established, while the corresponding gene expression (mRNA) signal may have already subsided. The new data concerning the inhibitory effect of Sol-DMAP on P-gp expression have been added to the manuscript (please see p. 7, lines 202-210).

You used one hour time point for tumor analysis, could you clarify why?

Authors: The accumulation of Sol-DMAP in tumor tissue (Fig. 2D) was assessed 1 hour after administration. This time point was chosen because, according to our obtained pharmacokinetic data, Sol-DMAP reaches its peak concentration in blood within this range (30 min – 1 h), enabling its detection in well-vascularized tissues such as the tumor mass. Similarly, the accumulation of doxorubicin (DOX) in tumor tissue was evaluated 1 hour after injection. This aligns with published data showing that DOX concentration in solid murine tumors typically peaks around this time point following administration [10].

Fig 4F legend: relative DCFDA level (lower partd) should be (lower part).

Authors: Corrected (please see p. 9, line 266).

All the results need to be easily read with  informative legend because it contains many data that is not easy to be followed, and also clarify symbols for example (+) in Fig 2D.

Authors: Corrected. Thank you for this important observation. All figure legends (except for the legend of Figure 1) have been significantly simplified for better reader comprehension. Please see p. 4, lines 127-130; p. 6, line 185; p. 9, lines 263-267; pp. 11-12, lines 296-311.

Discussion needed to be rearranged as follows: tumour geometry Nrf2 hypotheses and finally future research and commendation.

Authors: Corrected. In accordance with your suggestion, we have rearranged the Discussion section and added a paragraph describing the limitations of the current study, which can also be viewed as future prospects and recommendations (please see p. 13, lines 416-440; p. 14, lines 479-488).

We hope that corrected version of the manuscript will be acceptable for publication in the Pharmaceuticals.

Respectfully yours,

Dr. Andrey Markov

References

  1. Moralev, A.D.; Salomatina, O. V; Salakhutdinov, N.F.; Zenkova, M.A.; Markov, A. V Soloxolone N-3-(Dimethylamino)Propylamide Restores Drug Sensitivity of Tumor Cells with Multidrug-Resistant Phenotype via Inhibition of P-Glycoprotein Efflux Function. Molecules 2024, 29, 4939, doi:10.3390/molecules29204939.
  2. Lim, J.; Lee, H.; Hong, S.; Lee, J.; Kim, Y. Comparison of the Antioxidant Potency of Four Triterpenes of Centella Asiatica against Oxidative Stress. Antioxidants 2024, 13, 483, doi:10.3390/antiox13040483.
  3. Vasarri, M.; Bergonzi, M.C.; Leri, M.; Castellacci, R.; Bucciantini, M.; De Marchi, L.; Degl’Innocenti, D. Protective Effects of Oleanolic Acid on Human Keratinocytes: A Defense Against Exogenous Damage. Pharmaceuticals 2025, 18, 238, doi:10.3390/ph18020238.
  4. Lee, J.; Choi, M.K.; Song, I.S. Recent Advances in Doxorubicin Formulation to Enhance Pharmacokinetics and Tumor Targeting. Pharmaceuticals 2023, 16, 802, doi:10.3390/PH16060802.
  5. Bagdasaryan, A.A.; Chubarev, V.N.; Smolyarchuk, E.A.; Drozdov, V.N.; Krasnyuk, I.I.; Liu, J.; Fan, R.; Tse, E.; Shikh, E. V.; Sukocheva, O.A. Pharmacogenetics of Drug Metabolism: The Role of Gene Polymorphism in the Regulation of Doxorubicin Safety and Efficacy. Cancers (Basel). 2022, 14, doi:10.3390/CANCERS14215436.
  6. Lu, C.; Wei, J.; Gao, C.; Sun, M.; Dong, D.; Mu, Z. Molecular Signaling Pathways in Doxorubicin-Induced Nephrotoxicity and Potential Therapeutic Agents. Int. Immunopharmacol. 2025, 144.
  7. Wu, S.; Wang, W.; Dou, J.; Gong, L. Research Progress on the Protective Effects of Licorice-Derived 18β-Glycyrrhetinic Acid against Liver Injury. Acta Pharmacol. Sin. 2021, 42, 18–26, doi:10.1038/s41401-020-0383-9.
  8. Sarmah, D.; Sengupta, R. A Review on the Role of Phytoconstituents Chrysin on the Protective Effect on Liver and Kidney. Curr. Drug Discov. Technol. 2024, 21, doi:10.2174/0115701638242317231018144944.
  9. Wang, L.; Wei, C.; Jing, J.; Shao, M.; Wang, Z.; Wen, B.; Lu, M.; Jia, Z.; Zhang, Y. The Effects of Polyphenols on Doxorubicin-Induced Nephrotoxicity by Modulating Inflammatory Cytokines, Apoptosis, Oxidative Stress, and Oxidative DNA Damage. Phytother. Res. 2025, 39, 2147–2164, doi:10.1002/PTR.8470.
  10. Laginha, K.M.; Verwoert, S.; Charrois, G.J.R.; Allen, T.M. Determination of Doxorubicin Levels in Whole Tumor and Tumor Nuclei in Murine Breast Cancer Tumors. Clin. Cancer Res. 2005, 11, 6944–6949, doi:10.1158/1078-0432.CCR-05-0343.

Reviewer 2 Report

Comments and Suggestions for Authors

In this manuscript, Moralev et al. employ an integrated pharmacological, histological, biochemical, and computational approach to convincingly demonstrate that Sol-DMAP markedly enhances the antitumor efficacy of doxorubicin without increasing systemic toxicity. Furthermore, Sol-DMAP exhibits direct antitumor activity, effectively reducing tumor growth in vivo and inducing apoptosis and G1-phase arrest in RLS40 cells in vitro. In addition, Sol-DMAP attenuated DOX-induced hepatic injury, restored Hmox1 expression in liver tissue, and activated the NRF2-dependent antioxidant response in HepG2 cells. Collectively, the data suggest that Sol-DMAP may serve as a multifunctional therapeutic scaffold, combining P-gp inhibition with hepatoprotective properties. However, there are several comments and suggestions that should be addressed to further improve the manuscript.

Title:

The title is somewhat long and overly detailed. I recommend that the authors revise it to make it more concise and easier to understand while still reflecting the main findings of the study. A shorter, more focused title would enhance readability and better capture the reader’s attention.

Abstract:

I suggest that the authors revise the list of keywords to include more specific and relevant terms, such as lymphosarcoma, Sol-DMAP, and doxorubicin, instead of broader terms like cancer and ABC transporter. Using these precise keywords will improve the visibility and discoverability of the article.

Introduction:

I recommend that the authors revise the sentence in lines 44–46, beginning with “Tumor cells acquire resistance…”, to improve clarity and readability.

I suggest that the authors clarify whether Figure 1 was created by the authors or adapted from a previously published source. If the figure was adapted, please provide the appropriate references.

Material and methods:

I recommend that the authors include the data on animal body weights in the supplementary materials to provide additional context for the in vivo experiments. In addition, please provide a detailed primer information table in accordance with the MIQE guidelines, including the accession number, amplicon length, exon location, primer concentration, and amplification efficiency in the supplementary materials.

Results:

1- I suggest that the authors perform a Western blot analysis of P-gp in tumor tissues (Figure 3D) to assess whether Sol-DMAP affects P-gp expression at the post-translational level. This additional experiment would provide valuable mechanistic insight into the inhibitory effect of Sol-DMAP on P-gp.

2- To provide important insight into the selectivity and safety profile of this novel inhibitor. I recommend that the authors assess the safety of Sol-DMAP in normal cells, such as lymphocytes and hepatocytes (e.g., AML12), to evaluate its potential cytotoxic and ROS effects on non-cancerous cells. 

3- I suggest that the authors provide additional information regarding the effects of DOX and DOX + Sol-DMAP on ROS production and cell cycle arrest in Figures 4E and 4F. Alternatively, if these data have already been reported in a previous publication, please provide an appropriate reference and justify their inclusion in the text.

4- I kindly request the authors to clarify why all experiments in Figure 4 were performed with a 24-hour treatment, whereas the caspase activity assay was conducted after 6 hours. Please provide a justification for this difference in experimental timing.

5- I recommend that the authors provide the full names of CDDO, CDDO-Me, and CDDO-Im alongside their abbreviations in line 332 to improve clarity and facilitate better understanding of the context.

6- I kindly request the authors to provide a justification for how the combination of Sol-DMAP + DOX significantly activates HMOX1, GCLC, GCLM, and NQO1 genes in HepG2 cells, specifically compared to both untreated controls and DOX-treated cells.

Discussion:

I suggest that the authors include a discussion of the study’s limitations, particularly that the investigation was conducted only in combination with doxorubicin and not with other chemotherapeutic agents. Given the excellent quality of the work and the understandable constraints of time and resources for additional requested experiments, the authors may also justify these points as future perspectives and acknowledge them as limitations of the study.

Overall, I found this manuscript to be well-written, scientifically sound, and of significant interest. The authors present a comprehensive study investigating a novel P-gp inhibitor, Sol-DMAP, both in vivo (lymphosarcoma-bearing mice) and in vitro (RLS40 and HepG2 cells). Therefore, I would like to sincerely congratulate the authors for conducting a well-designed and comprehensive study.

Author Response

Dear Reviewer #2,

Thank you for taking the time to read and thoroughly analyze our article. We greatly appreciate your positive reception of our research, as well as your invaluable feedback. Addressing your insightful comments and suggestions during the revision has significantly strengthened our work.

We have revised the manuscript accordingly and provide our point-by-point responses below.

The title is somewhat long and overly detailed. I recommend that the authors revise it to make it more concise and easier to understand while still reflecting the main findings of the study. A shorter, more focused title would enhance readability and better capture the reader’s attention.

Authors: Corrected. Indeed, the original version of the title was cumbersome and difficult for readers to understand. Its shortened form retains references to the article's main points and, in our view, is more readable (please see p. 1, lines 2-4). Thank you for this important note.

I suggest that the authors revise the list of keywords to include more specific and relevant terms, such as lymphosarcoma, Sol-DMAP, and doxorubicin, instead of broader terms like cancer and ABC transporter. Using these precise keywords will improve the visibility and discoverability of the article.

Authors: Corrected. We sincerely appreciate this suggestion. Selecting keywords is a critical step, as it directly impacts the article's future visibility. The keyword list has been revised in accordance with your comment (please see p. 1, lines 36-37).

I recommend that the authors revise the sentence in lines 44–46, beginning with “Tumor cells acquire resistance…”, to improve clarity and readability.

Authors: Corrected. Please see p. 2, lines 42-44.

I suggest that the authors clarify whether Figure 1 was created by the authors or adapted from a previously published source. If the figure was adapted, please provide the appropriate references.

Authors: Addressed. We thank the reviewer #2 for the comment. The structures of the molecules depicted in Figure 1 and Figure 5B were created by the authors using ChemDraw molecule editor.

I recommend that the authors include the data on animal body weights in the supplementary materials to provide additional context for the in vivo experiments. In addition, please provide a detailed primer information table in accordance with the MIQE guidelines, including the accession number, amplicon length, exon location, primer concentration, and amplification efficiency in the supplementary materials.

Authors: Corrected. The table detailing animal body weights has been added to the supplementary material as Table S1 (referenced in the manuscript text on p. 5, line 167). In accordance with the MIQE guidelines, we have provided comprehensive primer information in the Supplementary Materials. Table S3 now includes the gene accession number, amplicon length, exon location, primer concentration, and amplification efficiency for all primers and probes used in the study.

1- I suggest that the authors perform a Western blot analysis of P-gp in tumor tissues (Figure 3D) to assess whether Sol-DMAP affects P-gp expression at the post-translational level. This additional experiment would provide valuable mechanistic insight into the inhibitory effect of Sol-DMAP on P-gp.

Authors: Corrected. Indeed, investigating the effect of Sol-DMAP on P-gp expression in tumor tissue at the protein level would significantly strengthen the mechanistic aspect of our work. In accordance with this suggestion, we performed an additional semiquantitative assessment of P-gp staining intensity in tumor sections using the following scale: 0 – none, 1 – mild, 2 – moderate, 3 – severe, and 4 – total.

The results of this analysis allowed us to conclude that Sol-DMAP statistically significantly suppresses P-gp expression at the protein level in tumor tissue (Fig. 3D). These new data concerning the inhibitory effect of Sol-DMAP on P-gp expression have been added to the manuscript (please see p. 7, lines 202-210). The results indicate that Sol-DMAP, a known direct inhibitor of P-gp [1], can also inhibit the expression of this protein during later stages of RLS40 lymphosarcoma development. This finding significantly expands our understanding of its P-gp-targeting potency.

2- To provide important insight into the selectivity and safety profile of this novel inhibitor. I recommend that the authors assess the safety of Sol-DMAP in normal cells, such as lymphocytes and hepatocytes (e.g., AML12), to evaluate its potential cytotoxic and ROS effects on non-cancerous cells. 

Authors: Corrected. We thank the reviewer #2 for this insightful question. We agree that assessing the effects of Sol-DMAP on non-cancerous cell types would further clarify its selectivity and safety profile. To test this, we performed additional cytotoxicity assays. Sol-DMAP exhibited lower cytotoxicity toward several normal cell models, namely human embryonic kidney HEK293 cells, mouse macrophages J774 cells, and Madin-Darby canine kidney MDCK cells (IC₅₀ values 2.3–4.0 μM) compared to DOX (IC₅₀ values 11.7–18.4 μM), suggesting an improved safety in vitro (please, see Fig. S1A, lines 217-224). Moreover, Sol-DMAP was significantly more effective than DOX in P-gp-overexpressing, RLS40 cells (IC₅₀ values 35.2 and 131.8 μM, respectively) and did not induce detectable toxicity in vivo, as evidenced by stable body weight, unchanged organ indices, and normal hepatic histology. At the same time, we recognize that despite its reduced toxicity relative to DOX, Sol-DMAP still requires further optimization to improve its selectivity toward tumor cells. According to ROS effects, Sol-DMAP increased ROS production by 1.5-fold in J774 macrophages (please, see Figure S1B, C, lines 251-253), supporting the need for optimization to avoid potential side effects associated with oxidative stress.

3- I suggest that the authors provide additional information regarding the effects of DOX and DOX + Sol-DMAP on ROS production and cell cycle arrest in Figures 4E and 4F. Alternatively, if these data have already been reported in a previous publication, please provide an appropriate reference and justify their inclusion in the text.

Authors: Corrected. We thank the reviewer for pointing this out. Indeed, the effects of DOX and DOX + Sol-DMAP on ROS production and cell cycle arrest were not addressed in the first version of the manuscript, but we have now added the corresponding data to the text and figures. In contrast to the ROS-independent mechanism of Sol-DMAP, DOX markedly increased ROS level in RLS40 cells, whereas the combination of Sol-DMAP with DOX completely suppressed this ROS-inducing effect (please, see Fig. 4F, lines 248-250). Regarding the cell cycle, although DOX alone induced G2/M arrest, the combination treatment was dominated by the effect of Sol-DMAP, resulting in a shift of the cell cycle profile toward G1-phase arrest (please, see Fig. 4G, H, lines 275-277). These results provide a more complete understanding of the effects of DOX and DOX + Sol-DMAP on both ROS production and cell cycle distribution.

4- I kindly request the authors to clarify why all experiments in Figure 4 were performed with a 24-hour treatment, whereas the caspase activity assay was conducted after 6 hours. Please provide a justification for this difference in experimental timing.

Authors: Addressed. We appreciate the reviewer’s question regarding the difference in treatment duration between the assays shown in Figure 4 and the caspase activity experiment. The assays performed at 24 h (Annexin-FITC staining, cell-cycle profiling, and ROS detection) reflect downstream cellular events. In contrast, caspase-3/7 activity was measured after 6 h because these proteins are activated at an earlier stage of apoptosis, prior to phosphatidylserine externalization detected by Annexin-FITC [2,3]. Additionally, in our experiments, Sol-DMAP and the combination treatment induced a strong apoptotic response, and by 24 h most cells had already progressed to late apoptosis, therefore, to investigate early apoptotic events, we assessed caspase-3/7 activity at the shorter 6 h time point. This allowed us to evaluate apoptosis progression comprehensively by measuring early (caspase activation) and late (phosphatidylserine externalization, cell cycle changes, ROS accumulation) markers. We have revised the manuscript to clearly state this (please, see line 238).

5- I recommend that the authors provide the full names of CDDO, CDDO-Me, and CDDO-Im alongside their abbreviations in line 332 to improve clarity and facilitate better understanding of the context.

Authors: Corrected. We thank the reviewer for this helpful suggestion. We have now added the full names of CDDO, CDDO-Me, and CDDO-Im alongside their abbreviations to improve clarity and facilitate understanding (please, see lines 345-346). In addition, the chemical structures of these molecules have been included in Figure 5B to provide a clearer visual representation and enhance comprehension of the context.

6- I kindly request the authors to provide a justification for how the combination of Sol-DMAP + DOX significantly activates HMOX1, GCLC, GCLM, and NQO1 genes in HepG2 cells, specifically compared to both untreated controls and DOX-treated cells.

Authors: Corrected. We thank the reviewer for this insightful comment. To address this, we specifically tested the effect of Sol-DMAP and DOX on NQO1 expression in HepG2 cells. Our results indicate that DOX alone induces NQO1 expression, likely reflecting a compensatory antioxidant response, whereas Sol-DMAP alone was shown to activate basal antioxidant gene expression. Interestingly, in the combination treatment, Sol-DMAP attenuated the DOX-induced overexpression of NQO1, suggesting that the triterpenoid mitigates the oxidative stress triggered by DOX (please, see Fig. 5G, H, lines 364-373). We note, however, that a comprehensive understanding of the combined effects on other NRF2-dependent genes (e.g. HMOX1, GCLC, GCLM) and the complete NRF2 signaling pathway would require further investigation. This point has been highlighted in the limitations section (please, see lines 480-482).

I suggest that the authors include a discussion of the study’s limitations <…>

Authors: Corrected. We are grateful for this suggestion. A paragraph addressing the study's limitations, which can also be viewed as future perspectives and recommendations for further research into the bioactivity of Sol-DMAP, has been added to the Discussion (please see p. 14, lines 479-488).

Overall, I found this manuscript to be well-written, scientifically sound, and of significant interest. The authors present a comprehensive study investigating a novel P-gp inhibitor, Sol-DMAP, both in vivo (lymphosarcoma-bearing mice) and in vitro (RLS40 and HepG2 cells). Therefore, I would like to sincerely congratulate the authors for conducting a well-designed and comprehensive study.

Authors: Dear reviewer #2, we are sincerely grateful for your positive and thoughtful review of our research. We hope this work will be valuable to researchers in the fields of triterpenoids, medicinal chemistry, and anticancer drug development, and that it may serve as a foundation for future collaborations with laboratories worldwide.

We hope that this version of the manuscript will be acceptable for publication.

Thank you very much!

Sincerely,

Dr. Andrey Markov

Akademgorodok, Russia

References

  1. Moralev, A.D.; Salomatina, O. V; Salakhutdinov, N.F.; Zenkova, M.A.; Markov, A. V Soloxolone N-3-(Dimethylamino)Propylamide Restores Drug Sensitivity of Tumor Cells with Multidrug-Resistant Phenotype via Inhibition of P-Glycoprotein Efflux Function. Molecules 2024, 29, 4939, doi:10.3390/molecules29204939.
  2. Mariño, G.; Kroemer, G. Mechanisms of Apoptotic Phosphatidylserine Exposure. Cell Res. 2013, 23, 1247–1248.
  3. Alshiraihi, I.; Kato, T.A. Apoptosis Detection Assays. In Chromosome Analysis: Methods and Protocols; Gotoh, E., Ed.; Springer US: New York, NY, 2023; pp. 53–63 ISBN 978-1-0716-2433-3.

Round 2

Reviewer 1 Report

Comments and Suggestions for Authors

The modified version is now clear, and the author has addressed the comments.

Reviewer 2 Report

Comments and Suggestions for Authors

All of my comments have been addressed thoroughly and satisfactorily by the authors. The revisions have significantly improved the clarity, scientific rigor, and overall quality of the manuscript. I am pleased with the authors’ thoughtful and comprehensive responses, and I commend them for their diligent work throughout the revision process.

Therefore, I am happy to recommend this manuscript for acceptance. I once again congratulate the authors on an excellent and well-executed study.